# Large-Emitting-Area Quantum Dot Light-Emitting Diodes Fabricated by an All-Solution Process

**DOI:** 10.3390/ijms241814350

**Published:** 2023-09-20

**Authors:** Ning Tu, S. W. Ricky Lee

**Affiliations:** 1Department of Mechanical and Aerospace Engineering, The Hong Kong University of Science and Technology, Hong Kong SAR, China; ntu@connect.ust.hk; 2Foshan Research Institute for Smart Manufacturing, The Hong Kong University of Science and Technology, Hong Kong SAR, China; 3Electronic Packaging Laboratory, The Hong Kong University of Science and Technology, Hong Kong SAR, China; 4Smart Manufacturing Thrust, The Hong Kong University of Science and Technology (Guangzhou), Guangzhou 511458, China; 5HKUST Shenzhen-Hong Kong Collaborative Innovation Research Institute, Shenzhen 518045, China; 6HKUST LED-FPD Technology R&D Centre, Foshan 528200, China

**Keywords:** quantum dots, light-emitting diode, materials, photophysical

## Abstract

Quantum dots (QDs) have attracted a lot of attention over the past decades due to their sharp emission spectrum and color, which can be tuned by changing just the particle size and chromophoric stability. All these advantages of QDs make quantum dot light-emitting diodes (QLEDs) promising candidates for display and light-source applications. This paper demonstrates a large-emitting-area QLED fabricated by a full-solution process. This QLED is composed of indium tin oxide (ITO) as the anode, poly(3,4-ethylenedioxythiophene)-poly(styrenesulfonate) (PEDOT: PSS) as the hole injection layer (HIL), and poly(*N*,*N*′-bis-4-butylphenyl-*N*,*N*′-bisphenyl)benzidine (poly-TPD) as the hole-transport layer (HTL). The light-emitting layer (EML) is composed of green CdSe/ZnS quantum dots. By applying the ZnO nanoparticles as the electron-injection/transport layer, QLED devices are prepared under a full-solution process. The large-emitting-area QLED exhibits a low turn-on voltage of around 2~3 V, and the International Commission on Illumination (CIE) 1931 coordinate value of the emission spectrum was (0.31, 0.66). The large emitting area and the unique QLED structure of the device make it possible to apply these features to inkjet printing quantum dot light sources and quantum dot display applications.

## 1. Introduction

Colloidal quantum dots (QDs) are solution-growth semiconductor nanocrystals. Due to the quantum confinement effect, the discrete energy level of QDs gives rise to atomic-like emission with narrow linewidths. The emission color of QDs can be tuned by changing the size of QDs and their composition, thus providing the emission wavelength from ultraviolet wavelengths to near-infrared wavelengths [1,2,3,4,5]. Capitalizing on the luminous properties of QDs, such as wide color emission spectrum ranges, a narrow emission spectrum with a small value of full-width half maxima (FWHM), and inherent photo stability, there has been significant growth in the commercial applications of QLEDs. The first application of a quantum dot light-emitting diode (QLED) device was first published in 1994 by V. L. Colvin [6]. The QLED device structure had only 100 nm thick PPV as the hole-transport layers, while five layers of QDs worked not only as the electron transport layer but also as the color emission layer. Meanwhile, ITO performed as the anode, and Mg coated with Ag performed as the cathode. The electroluminescence spectrum was broadened with the effect of the PPV emission peak, but the external quantum efficiencies were still low, at 0.001–0.1%. Over the last few years, QLEDs have received much attention both from academic research and commercial industries, with a great deal of effort devoted to improving their efficiency, color purity, and fabrication flexibility. 

After 8 years, the luminescence efficiency of QLEDs has been improved 25-fold by Coe-Sullivan et al., through sandwiching a single QD between two organic thin films. This type of QLED amalgamates the diversity of organic materials with the high-performance electronic and optical properties of QDs. The holes are injected from the ITO contact interface into the poly-TPD layer, and then transported to a single QD monolayer, while electrons are injected from the Mg: Ag cathode into the Alq_3_ and move to the QD monolayer. Thus, the excitation of the organic molecules is achieved through exciton energy transfer and direct charge injection, and these parallel processes occur simultaneously in the QD layer [7]. However, organic electron-injection/transport layers are sensitive to moisture and temperature. Compared to deciding on organic electron-injection materials, choosing suitable materials for the inorganic electron-injection/transport layers is among the most crucial decisions for the development of QLED devices [8]. MoS_2_, NiO, and TiO_2_ have been applied as inorganic charge-transport layers. However, this type of monocrystalline material needs to be processed at high temperatures, which affects a wide range of applications [9,10,11]. Alternatively, ZnO nanoparticles are a more stable and reliable choice compared to other inorganic charge-transport layers, especially in the application of electron-injection/transport layers. Kwak et al. demonstrated the making of red, green, and blue QLEDs by applying ZnO nanoparticles as the electron-injection/transport layers. The QLED structure is an inverted device structure, while electrons are injected from ITO, by optimizing the energy level of the other organic hole-transport layers. The best device performance has been demonstrated in a structure with 4,4′-bis(carbazole-9-yl) biphenyl (CBP) as the hole-transport layer [12]. Pan et al. also synthesized ZnO nanoparticles through a sol–gel method with different diameters. It has been shown that reducing the diameter of the ZnO nanoparticles results in a 60-fold increase in QLED performance. This is because ZnO nanoparticles with a smaller diameter have higher spatial confinement, and the tendency towards band-gap enlargement is consistent with decreasing ZnO nanoparticle size [13]. In particular, ZnO nanoparticles’ higher electron mobility improves the charge injection efficiency. Furthermore, the band offset in ZnO also contributes to the hole confinement in the emission layer, thereby boosting the charge recombination efficiency of QLEDs [8,14,15]. Moyen et al. have shown that the defect density also decreases with decreasing ZnO nanoparticle size. Therefore, reducing nanoparticles’ size not only prevents exciton quenching in the charge-transport layer and emission-layer interface but also improves the charge balance in QLEDs [16,17]. 

Despite this recent progress, there are still challenges that remain in applications of QLEDs in relation to displays or solid-state lighting, such as the poor confinement of excitons in large-area QLEDs by the solution-processable method [18,19,20]. The functional layers in QLEDs are fabricated by a spin-coating process. After the functional material is deposited on the desired substrate, there will be a wetting and drying process. The solvent will evaporate simultaneously, while the material will remain on the substrate. In the meantime, the performance of QLEDs is closely dependent on the charge transport efficiency and recombination of electrons and holes. If the functional layer underneath is partially removed, it will cause shorting and non-emissive problems in QLEDs [21,22,23]. Thus, solvent selection is important during the fabrication of QLEDs, because the solvent needs to protect the functional layers underneath and help the effective deposition of the next functional layer. Furthermore, the uniformity of the QD film also affects the emission properties of QLED devices, which is especially important for large-emitting-area QLED devices. A smooth and pinhole-free charge-transport layer is needed underneath to ensure that the above QD emission layer is extremely smooth. In common QLED structures, PEDOT: PSS is often spin coated onto the ITO surface to smooth the ITO surface and obtain a stable deep work function. As part of the continuous development to increase the conductivity of PEDOT: PSS, the PEDOT: PSS can now replace ITO to work as the anode. Conjugated polymers such as poly{*N*,*N*′-bis(4-butylphenyl)-*N*,*N*′-bis(phenyl)-benzidine} (poly-TPD) [24,25,26], poly-(9-vinylcarbazole) (PVK) [25,26], and poly[(9,9-dioctylfluorenyl-2,7-diyl)-alt-(4,4′-(N-(4-butylphenyl) (TFB) [27] dissolve in chlorobenzene, and consequently are coated on PEDOT: PSS to act as a hole-transport layer and electron-blocking layers. Nevertheless, chlorobenzene is also commonly used for QD dispersion. Therefore, if we use chlorobenzene or another nonpolar organic solvent to disperse QDs, this will cause the hole-transport layer to suffer re-dissolving. Meanwhile, a similar problem can arise in the deposition electron-transport layer; for example, ZnO nanoparticles are normally dispersed in a polar solvent, as this solvent is not compatible with nonpolar organic solvents, and the combination will affect the wetting properties of the electron-transport layer, which further affects the electron-transport layer completely covering the emission layer. These problems can lead to electron leakage and non-emissive recombination problems. 

Herein, we introduce easily processable and well-controlled large-emitting-area QLEDs by an all-solution process deposition method using the ZnO NPs solution as the electron-transporting layer and electron-injection layer. This study demonstrates the integration of ZnO nanoparticles into the QLED by controlling the fabrication process, and the promising rich electron transporter. By choosing a suitable solvent with compatible surface tension and saturated vapor pressure, the functional layers will be well-prepared and the film morphology of the emission layer improved. The reproducibility and unique device structure suggest that this technique could be widely applicable to the fabrication of other high-quality large-emitting-area organic/inorganic devices.

## 2. Results and Discussion

The structure and schematic energy band diagram of the QLED are illustrated in Figure 1a,b respectively. The ITO was prepared as the anode. The PEDOT: PSS was used as the hole-injection layer, while poly-TPD was applied as the hole-transport layer, and the ZnO nanoparticles were used as the electron-injection/transport layer. The green (CdSeS/ZnS) QDs were prepared as the color-emission layer. Al was deposited on the ZnO nanoparticle layer as the cathode. The film thickness is illustrated in Appendix A. According to Figure 1b, the high energy bandgap between PEDOT: PSS and QDs made it difficult for the holes to be transported to the emission layer. Thus, poly-TPD was introduced as the hole-transport layer to decrease the energy barrier between PEDOT: PSS and the QD layer, which increased the number of holes injected into the emission layer. Meanwhile, the ZnO nanoparticles reduced the conduction band offset at the QD/ZnO nanoparticle interface, stemming from the ~4.0 eV electron affinity. Meanwhile, the ~7.3 eV ionization potential of ZnO nanoparticles also induced a large valence band offset to restrict the holes within the emission layer. Thus, the wide energy bandgap of the ZnO nanoparticle layer not only improved the electron injection from the cathode into the emission layer but also prevented the leakage of holes to the ZnO nanoparticle layer. This restricted the excitation-recombination region and prevented exciton quenching in the poly-TPD/QD interface, hence potentially improving the charge recombination efficiency [13,16,17,28,29,30,31]. 

Transmission electron microscopy (TEM) images of the ZnO nanoparticle samples are illustrated in Figure 2a,b. According to Figure 2a, the ZnO nanoparticles’ diameter is 3 nm on average with an ellipsoidal shape. Meanwhile, the lattice fringes can be observed in Figure 2b, which suggests good crystallinity of ZnO nanoparticles. Pan et al. found that the tendency for ZnO nanoparticle bandgap enlargement corresponded with the decrease in ZnO nanoparticles’ size, which was consistent with the relation based on effective mass approximation [32]. However, Moyen et al. found that the effective mass approximation model overestimated the blue shift of ZnO once the diameter of ZnO nanoparticles widened. Viswanatha et al. applied a more realistic tight-binding model to reproduce the variations of the ZnO bandgap with the following expression [16,17,33]:
$$E_{g,\;NPs}=E_{g,bulk}+\frac{100}{18.1d^{2}+41.4d-0.8\;}$$
where $E_{g,NPs}$ is the bandgap energy of the nanoparticles, $E_{g,bulk}$ is the bandgap energy of bulk ZnO (~3.3 eV) [34], and d is the diameter of nanoparticles. The results were double confirmed in Moyen et al.’s research.

The performance of the device is summarized in Figure 3a,b. It can be seen in Figure 3a that the current density versus voltage (I–V) curve illustrates the low turn-on voltage of the device, of around 3 V. The low turn-on voltage is due to the high mobility of ZnO nanoparticles. The size of ZnO nanoparticles applied in the QLED device was only 3 nm, which resulted in a wide energy bandgap. This widened energy bandgap of ZnO nanoparticles worked with the design of the QLED structure to enable the electrons to be easily injected into the emission layer. This was due to the high mobility of ZnO nanoparticles, accompanied by the low energy barrier between the cathode and emission layer. In the meantime, the high energy barrier between the emission layer and hole-transport layer made the electrons accumulate at the interface of the QDs and poly-TPD. By analogy, the holes accumulated at the interface of the emission layer and electron-injection/transport layer. The Auger-assisted hole-injection process forces the high-energy holes to cross the injection barrier layer and recombine with electrons inside the emission layer and emit photons [25,26,35]. The electroluminescence (EL) spectrums under different voltages can be observed in Figure 3b. According to Figure 3b, the intensity of the electroluminance (EL) spectrums increases with the increase in applied voltage. The emission wavelength of electroluminance spectrums was 534 nm, which was 4 nm red-shifted compared to the spectrum of the green QDs solution. The properties of green QDs solution is Appendix A. The full-width half maximum (FWHM) of electroluminance (EL) spectrums enlarged to 44 nm, which might be affected by oxygen and moisture, because the device was prepared without protection by inert gas. The color coordinate value according to CIE 1931 is illustrated in Figure 4a, while the color coordinate value was (0.31, 0.66) with a photometric value of 0.11 lm. Figure 4b shows a lit-up QLED device and a 6 × 6 mm^2^ square area emitting uniform green light. The measurement setup equipment of QLED is shown in Figure 4c. The integrating sphere was applied to measure the optical properties of the QLED device with a Keithley source meter to provide electric power to the QLED device. 

## 3. Materials and Methods

### 3.1. ZnO Nanoparticle Synthesis

The ZnO nanoparticles were synthesized by a modified nucleation dissolution recrystallization growth method, which is schematized in Figure 5. Potassium hydroxide (KOH) solution was prepared by dissolving 13 mmol KOH in 16 mL of methyl alcohol (CH_3_OH), while 5.4 mmol zinc acetate dehydrate (ZnC_4_H_6_O_4_) was dissolved in 31 mL of CH_3_OH at 60 °C under continuous stirring. The pre-prepared KOH solution was then fed into zinc acetate solution with a feeding rate of 0.8 mL/min. The solution was stirred at 800 rpm and heated at 60 °C for 1.7 h. The heater and stirrer were then removed to allow the solution to sit for another 2 h. When the ZnO nanoparticles had settled in the bottle of the flask, the mother solution was removed and the ZnO nanoparticles were washed twice with CH_3_OH with a centrifuge. The washed ZnO was dispersed in CH_3_OH for further use. 

### 3.2. QLED Preparation 

The substrates were cleaned by detergent, CH_3_OH, acetone, and isopropyl alcohol (IPA) consecutively, each for 20 min under ultrasonic conditions, followed by ozone plasma treatment for 30 min to modify the surface tension of ITO glass. The PEDOT: PSS (CLEVIOS, PH1000) was spin-coated on the cleaned ITO surface at 3000 rpm for 30 s, while the PEDOT: PSS film was baked at 150 °C for 30 min under vacuum protection. The poly-TPD solution was prepared by dissolving 0.5 wt.% poly-TPD in chlorobenzene and spin-coated onto the baked PEDOT: PSS surface at 3000 rpm for 60 s and annealed at 100 °C for 30 min under a vacuum. The QDs were dispersed in hexane at a concentration of 13 mg/mL and spin-coated on a poly-TPD surface at 3000 rpm for 30 s and baked at 80 °C for 30 min under a vacuum. The ZnO nanoparticle solution was prepared by dissolving 35 mg/mL ZnO nanoparticles in chloroform and methyl alcohol mixed solvent at the volume ratio of 1:9. Subsequently, the ZnO nanoparticles were spin-coated at 1500 rpm for 60 s and annealed at 60 °C for 30 min. Then, 100 nm Al was deposited on the ZnO nanoparticle layer by the vapor-deposition method (Denton Vacuum Explorer 14). 

The film thickness was measured by the Alpha-Step 200 Tencor surface profiler. The current–voltage (I–V) curve was measured by a Keithley 2612A system source meter, and a CAS 140CT-151 array spectrometer was applied to measure the photoluminescence. The electroluminance spectrum was measured by an ISP 500–100 Integrating sphere. The CIE chromaticity coordinates were measured according to the electroluminance spectra. All the measurements were taken at room temperature without encapsulation. The structure of nanoparticles was characterized by a Cs-corrected high-resolution transmission electron microscope (HRTEM, JEM 2010). The green CdS/ZnS QDs were purchased from Mesolight Inc. (Suzhou, China). 

## 4. Conclusions

In this paper, the nucleation dissolution recrystallization growth method was presented for the synthesis of ZnO nanoparticles. The synthesized ZnO nanoparticles were applied on the electron-injection/transport layer to fabricate a large-emitting-area QLED device. The small diameter of ZnO nanoparticles demonstrated high electron mobility and a wide energy bandgap, which improved the charge recombination efficiency of the QLED device. The solvent of ZnO nanoparticles was mixed with an appropriate orthogonal solvent to prevent the dissolving of existing layers. This method helped ensure that high-quality and compact ZnO nanoparticles formed a film. Finally, the generation of a large-emitting-area QLED based on ZnO nanoparticles, mixed with an appropriate dispersing solvent, was demonstrated in this paper. Moreover, by capitalizing on the advantages of all-solution-processed large-emitting-area QLEDs, there is great potential for the fabrication of quantum dot light sources and quantum dot displays via the printing method in the consumer market in the future.

## Figures and Tables

**Figure 1 ijms-24-14350-f001:**
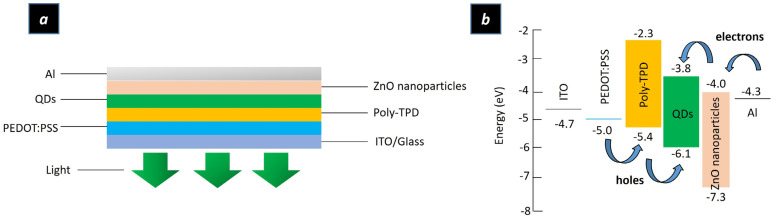
(**a**) Structure of the QLED device (**b**) energy diagram of the QLED device.

**Figure 2 ijms-24-14350-f002:**
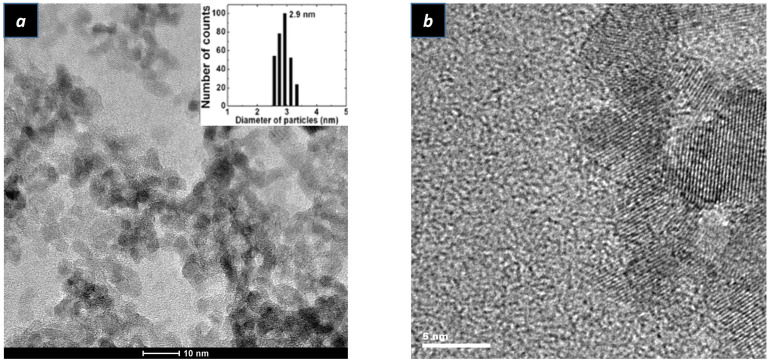
(**a**) TEM image of ZnO nanoparticles. Insert image is the statistical distribution images of particle size (**b**) HRTEM image of ZnO nanoparticles.

**Figure 3 ijms-24-14350-f003:**
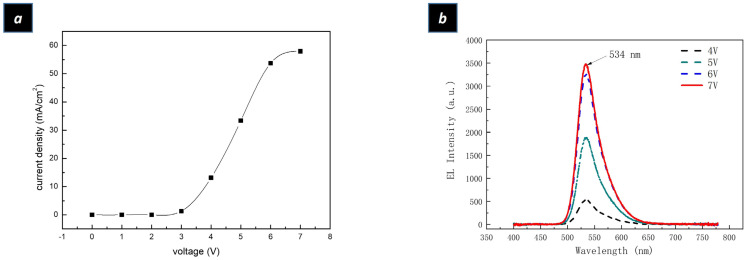
(**a**) Current density–voltage characteristics of a QLED device (**b**) EL spectrums of a QLED device with different voltages.

**Figure 4 ijms-24-14350-f004:**
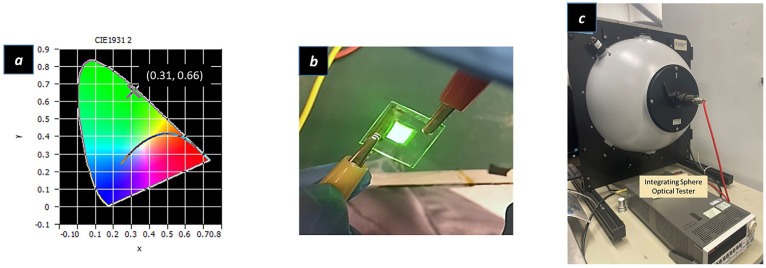
(**a**) 1931 CIE coordinate value of a QLED device, (**b**) Lit-up QLED device, (**c**) QLED device measurement setup equipment.

**Figure 5 ijms-24-14350-f005:**
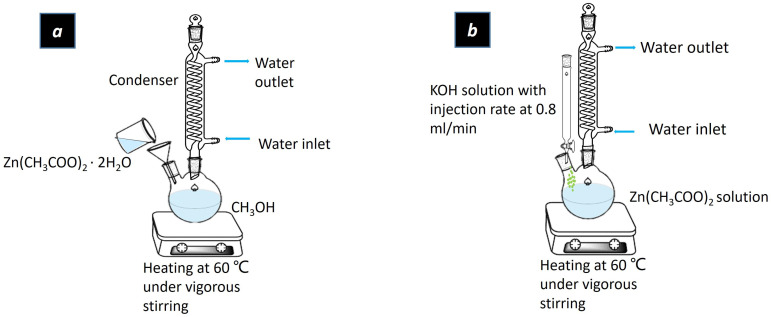
Schematic representation of the fabrication process of ZnO nanoparticles (**a**) zinc acetate solution preparation (**b**) the reaction process of ZnO nanoparticles.

## Data Availability

The data presented in this study are available upon request from the corresponding authors.

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
