# Peer review of "Large-Emitting-Area Quantum Dot Light-Emitting Diodes Fabricated by an All-Solution Process"

_ijms, 2023, doi:10.3390/ijms241814350_

Round 1

Reviewer 1 Report

This work looks interesting, but it needs a few questions to be resolved before publication as given below:

1.       The figure's quality is very low. Please improve it.

2.       What is the problem statement of your work?

3.       Labeling in Figure 2a seems stretched.

4.       Authors directly used EL. At least once write Electroluminescence (EL) in the manuscript.

5.       Please show the original optical images of the devices.

6.       It is better to demonstrate the measurement setup.

7.       To make the introduction broader please add a few recent articles:

a.       doi: https://doi.org/10.1016/j.cej.2023.143923

b.       Applied Optics, 62(17), 4431-4438. doi: 10.1364/AO.491732

c.       Materials 2022, 15(9), 3364; https://doi.org/10.3390/ma15093364

d.       Materials Letters, 350, 134868. doi: https://doi.org/10.1016/j.matlet.2023.134868

8.       The information from TEM is very poor. Please add phase profile from FFT profile for lattice direction of ZnO.

9.       Authors claim pin-hole-free film but there is no evidence.

10.   I should recommend adding scientific logic/reasons for your results/outcomes of this manuscript.

English should be improved.

Reviewer 2 Report

The Authors submitted a report on manufactiring terchnology of quantum dot light-emitting diode (QLED). The results are interesting; however, there are several issues to fix or explain prior discussion on the manuscript acceptance.

#1 Material solubility

Since the all-solution process of multilayer structures require spin-coating of several layers the solubility of materials is important. The PEDOT:PSS is water-based solution and therefore the poly-TPD in chlorobencene can be easily spun on the top (I am not sure about the wetting properties, but I shall trust to the Authors). The QD nanoparticles are dissolved in octane and spun on poly-TPD. The Auhots should provide a proof that the poly-TPD is not soluble in octane. On the top of QD nanoparticles are spun ZnO nanoparticles in chloroform and methyl alcohol mixture. The Authors should provide a solid proof that the QD nanoparticle layer is not dissolved by the ZnO nanoparticle solution.

#2 Layer thickness credibility and reliability

The Authors claim that the ZnO layer is only 5 nm thick. If we assume the nanoparticel sice of about 2.9 nm (as stated by the Authors) it represents less than two layers of nanopartcles. Since the spin-coating cannot fabricate homogeneous layers of such thickness the Authors should provide a solid proof of their statements. The measure,ents of surface profiles at such low thickness (and probably not homogeneous and continuous layer) id not a proof at all.

#3 Material and technology informations are missing or are missleading

There is a littl etypoe, the PEDOT:PSS is not "H1000" but "PH1000". Important information is that it is a water-based solution. As a result, after spin-coating there is still a wet surface (it is not volatile solvent). Hence, the vacuum drying is improper and may cause additional unintentional effects. Futhermore, there is no information on quantum dot nanoparticle source, if the QD nanoparticels were synthesized by Authors or fabricated by industrial partner.

As a result, there are serious issues with submitted manuscript and I cannt support it even though it is interesting. Hence, I have no choice but to require major revisions with additional explanations and experiments.

No comments on English level. There are some typos; however, the scientific issue is worse.

Reviewer 3 Report

After carefully review the manuscript entitled: "Large Emitting Area Quantum-Dots Light Emitting Diodes Fab- 2 ricated by All-Solution Process" I can mention that the paper is very interesting from the point of view to know about how to increase the area of the fabricated devices, which commonly show many issues about the carrier transporting layers and the carrier flow through the devices, taking into account the large dimensions (higher resistance to carrier transport). I can recommend this paper in the actual form. Some english errors (minor) can be addressed along the text.

Minor english errors can be addressed along the main text

Author Response

Thank you very much for taking the time to review this manuscript.

The English errors have been addressed in the text. Please find the detailed responses in the corresponding revisions in track changes in the re-submitted files.

Round 2

Reviewer 1 Report

In the revised manuscript authors replied to all the questions. I should recommend this manuscript for publication. But one minor issue must be resolved before publication.

1. Figure 5 still has a problem with labeling. 

 Minor editing of English language required

Author Response

Thank you very much for taking the time to review this manuscript. 

The labelling of Figure 5 have revised in the updated manuscript.

The English has minor editing in the updated manuscript.

Thank you for your time and kind consideration.

Reviewer 2 Report

The Authors answered all questions and/or comments of the Reviewer and the mansucript was fixed according to the response. As a result, I have no other option but to support the manusccript for the publication.

No specific comments on the English level.

Author Response

Thank you very much for taking the time to review this manuscript.